# LLMCO$_2$: Advancing Accurate Carbon Footprint Prediction for LLM Inferences

## Abstract

Throughout its lifecycle, a large language model (LLM) generates a substantially larger carbon footprint during inference than training. LLM inference requests vary in batch size, prompt length, and token generation number, while cloud providers employ different GPU types and quantities to meet diverse service-level objectives for accuracy and latency. It is crucial for both users and cloud providers to have a tool that quickly and accurately estimates the carbon impact of LLM inferences based on a combination of inference request and hardware configurations before execution. Estimating the carbon footprint of LLM inferences is more complex than training due to lower and highly variable model FLOPS utilization, rendering previous equation-based models inaccurate. Additionally, existing machine learning (ML) prediction methods either lack accuracy or demand extensive training data, as they inadequately handle the distinct prefill and decode phases, overlook hardware-specific features, and inefficiently sample uncommon inference configurations. We introduce LLMCO$_2$, a graph neural network (GNN)-based model that greatly improves the accuracy of LLM inference carbon footprint predictions compared to previous methods.

## 1 Introduction

Large language models (LLMs) (BigScience, 2023; Meta, 2024; Mistral, 2024) have demonstrated high efficacy across various generative Natural Language Processing (NLP) tasks, such as code completion (Nam et al., 2024), question-answering (Shao et al., 2023), and text summarization (Pilault et al., 2020). Their integration into daily activities (e.g., web browsing (Campello de Souza et al., 2023)) highlights their increasing prevalence. However, this widespread adoption has led to significant carbon dioxide equivalent (CO2eq) emissions (Luccioni et al., 2024). For instance, training the Google T5 LLM generates 40% more carbon emissions than a round-trip flight between San Francisco and New York (Faiz et al., 2024).

Inferences for LLMs can produce an even larger carbon footprint than their initial training. Conservative estimates suggest that OpenAI handles over 270 million daily inference requests (Chien et al., 2023), with an average prompt length of 1.2K tokens per request (Patel et al., 2024). Training GPT-4 requires approximately 13 trillion tokens (OpenAI, 2024), with a single epoch requiring three times the FLOPs of an inference (Faiz et al., 2024). Consequently, the carbon emissions from 121 days of serving GPT-4 inferences equate to those of its training. As the volume of daily inference requests rises, and with increased adoption of LLMs across various applications (OpenAI, 2024), the period required for inference emissions to match training emissions is rapidly decreasing.

Given the significant environmental impact, it is essential for both end users and cloud providers to understand the carbon emission costs of different service-level objectives (SLOs) (Patel et al., 2024) for LLM inference accuracy and latency. An accurate carbon footprint prediction tool is crucial before initiating inference requests, enabling users to assess trade-offs between accuracy, latency, and carbon emissions. This tool would also help cloud providers justify billing policies transparently, promoting low-carbon practices among users.

However, *there is a lack of modeling tools for accurately estimating the carbon footprint of LLM inferences*. Users submit LLM inference requests with varying configurations (e.g., batch size, prompt length, and token generation number) to cloud services, while cloud providers employ different GPU types and quantities to meet diverse SLOs for accuracy and latency. Prior studies (Nguyen et al.,

Figure 1: A decoder-only LLM's autoregressive inference consists of prefill and decode phases.

Figure 2: Comparing the energy of 2 phases in a Bloom-3B inference on an Nvidia L4 GPU.

2024; Luccioni et al., 2024) have reported LLM inference carbon emissions on a limited range of GPUs, but exhaustively profiling all possible configurations is impractical. Although a carbon footprint model for LLM training (Faiz et al., 2024) exists, its equation-based approach fails to accurately capture LLM inference emissions due to lower and highly variable model FLOPS utilization (MFU) (Pope et al., 2023). While machine learning (ML)-based tools have been developed to predict inference latency (Liu et al., 2023; Zhang et al., 2021) and energy (Tu et al., 2023) for neural networks on mobile devices, applying them to LLM inference carbon footprints results in low accuracy or requires extensive training data for the following reasons:

- *Lack of autoregressive consideration.* Existing tools treat CNN inference as a single task, failing to account for the distinct autoregressive phases of LLMs: the compute-intensive prefill and the memory-bound decode (Patel et al., 2024). This oversight reduces prediction accuracy or necessitates large training datasets.
- *Neglect of hardware-specific features.* Prior tools overlook critical hardware characteristics such as GPU peak computing throughput, memory bandwidth, and network bandwidth. Accurate predictions thus require exhaustive profiling across different GPU platforms (Pope et al., 2023), making these methods inefficient and less reliable for new hardware.
- *Disregard for prevalent configurations.* Previous tools treat all inference configurations equally, ignoring that most real-world LLM inference requests, such as those in Azure Cloud (Microsoft, 2024), feature small batch sizes, shorter prompts, and limited token generation. This approach fails to improve prediction accuracy for typical usage scenarios.

We introduce LLMCO$_2$, an accurate carbon footprint regression model for LLM inferences. LLMCO$_2$ employs a novel graph embedding technique that represents each transformer layer's kernels as a graph, with nodes indicating kernels and edges capturing data dependencies. Node features for the prefill and decode phases are encoded separately, incorporating each node's Roofline performance as a hardware-specific feature. We also develop a focused data sampling algorithm that emphasizes common configurations of inference requests, LLM architectures, and GPU setups. LLMCO$_2$ improves carbon footprint prediction accuracy by 51%-123% over existing ML-based energy predictors for LLMs across diverse inference requests and GPU configurations.

## 2 BACKGROUND

**Autoregressive LLM Inferences**. As shown in Figure 1, during autoregressive inference (Pope et al., 2023) of a decoder-only LLM, all input tokens are processed in parallel during the first iteration, generating the first token—this is the prefill phase (Patel et al., 2024). The context from the LLM's attention layers is stored in the key-value (KV) cache for future iterations. Subsequent tokens are then generated using the latest token and the KV cache as inputs, forming the decode phase (Patel et al., 2024).

**Distinct Characteristics of Two Phases**. In autoregressive LLM inference, the prefill phase computes and stores context in the KV cache, while the decode phase primarily accesses this cache. The prefill phase is compute-bound, relying on GPU cores, whereas the decode phase is memory-bound, relying on GPU memory. Consequently, the phases differ in latency, energy consumption, and carbon footprint. As shown in Figure 2, during a Bloom-3b inference (BigScience, 2023) with a batch size of 1 on an Nvidia L4 GPU, the prefill phase's energy use is negligible for requests with many generated tokens but dominates when fewer tokens are generated. Treating both phases as a single task without sufficient training data significantly reduces energy prediction accuracy.

**Kernels in a transformer Layer**. As illustrated in Figure 4, a transformer layer (Vaswani et al., 2017) comprises a masked multi-head attention (MHA) layer and a feed-forward (FF) layer framed by normalization layers. In the MHA layer, the attention mechanism is executed via $Q_{proj}$, $K_{proj}$, and $V_{proj}$ kernels. Techniques like Grouped-Query Attention (Ainslie et al., 2023) and Flash Atten-

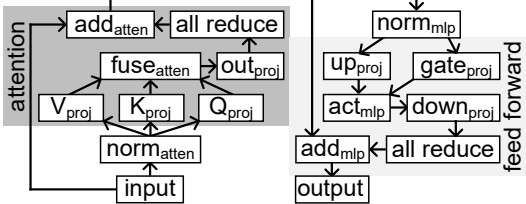

| GPU$_0$ | GPU$_1$ | GPU$_2$ | GPU$_3$ | GPU$_0$ | GPU$_1$ | GPU$_2$ | GPU$_3$ | GPU$_0$ | GPU$_1$ | GPU$_2$ | GPU$_3$ |
|---|---|---|---|---|---|---|---|---|---|---|---|
| $X_0^0$ | $X_0^1$ | $X_0^2$ | $X_0^3$ | $\Sigma_j X_0^j$ | | | | $\Sigma_j X_0^j$ | $\Sigma_j X_0^j$ | $\Sigma_j X_0^j$ | $\Sigma_j X_0^j$ |
| $X_1^0$ | $X_1^1$ | $X_1^2$ | $X_1^3$ | | $\Sigma_j X_1^j$ | | | $\Sigma_j X_1^j$ | $\Sigma_j X_1^j$ | $\Sigma_j X_1^j$ | $\Sigma_j X_1^j$ |
| $X_2^0$ | $X_2^1$ | $X_2^2$ | $X_2^3$ | | | $\Sigma_j X_2^j$ | | $\Sigma_j X_2^j$ | $\Sigma_j X_2^j$ | $\Sigma_j X_2^j$ | $\Sigma_j X_2^j$ |
| $X_3^0$ | $X_3^1$ | $X_3^2$ | $X_3^3$ | | | | $\Sigma_j X_3^j$ | $\Sigma_j X_3^j$ | $\Sigma_j X_3^j$ | $\Sigma_j X_3^j$ | $\Sigma_j X_3^j$ |
| (a) initial state | | | | (b) reduce-scatter | | | | (c) all-gather | | | |

Figure 3: An example of a all-reduce kernel running on four GPUs.

tion (Dao, 2024) reduce memory overhead in LLM inferences by employing fewer key-value heads and fusing attention computations ($fuse_{atten}$). The FF layer operates as a two-layer MLP. To enable tensor parallelism (Narayanan et al., 2021), two all-reduce kernels are employed to distribute General Matrix Multiply (GEMM) operations across GPUs.

**All-reduce**. Given the huge memory demands of LLMs and the limited capacity of individual GPUs, multiple GPUs connected via PCIe or NVLink (Patel et al., 2024) are crucial for LLM inferences. Tensor parallelism (Aminabadi et al., 2022) splits tensors across GPUs and replicates all layers, providing a significant speedup over other parallelism strategies. To support tensor parallelism, two all-reduce kernels are incorporated into each transformer layer. An all-reduce kernel (Hidayetoglu et al., 2024)

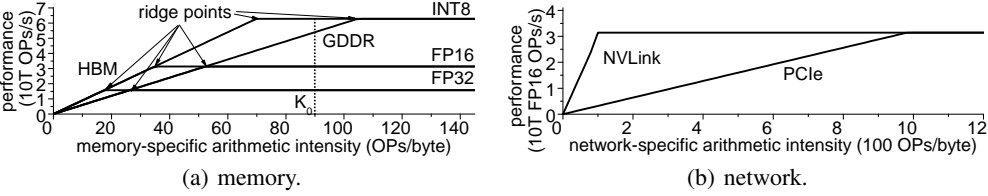

Figure 4: The kernels in a transformer layer.

consists of a reduce-scatter operation followed by an all-gather operation, as shown in Figure 3. For instance, a $4 \times 4$ matrix, evenly distributed across four GPUs (each holding a column), undergoes reduce-scatter, where each row is assembled and summed on one GPU, followed by all-gather, where the summed values are shared across all GPUs.

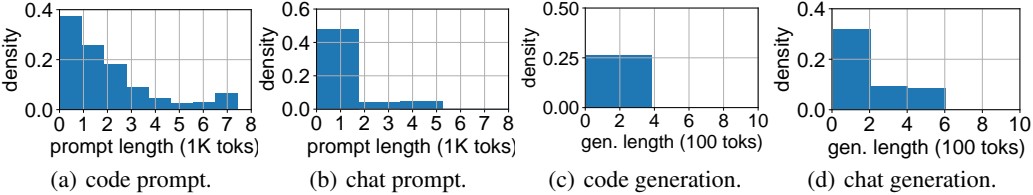

| (a) memory. | (b) network. |
|---|---|

Figure 5: The memory and network Roofline models for an Nvidia V100 GPU.

**Roofline model**. The Roofline model (Cardwell & Song, 2019) is a performance analysis tool that estimates a kernel's performance on a GPU, measured in operations per second (OPs/s). It considers factors like peak GPU throughput, peak memory and network bandwidth, and the kernel's memory- and network-specific arithmetic intensities. The memory and network Roofline models for an Nvidia V100 GPU (Nvidia, 2017) are shown in Figures 5(a) and 5(b), respectively. The X-axes represent arithmetic intensity, calculated as total kernel operations divided by total memory or network bytes transferred. A ridge point, or 'balance' point, indicates where compute and data movement performance meet. The Nvidia V100 supports FP32, FP16, and INT8 operations with HBM or GDDR memory, resulting in six ridge points in Figure 5(a). Two network interfaces (NVLink and PCIe) yield two ridge points in Figure 5(b). Kernels with arithmetic intensities below the ridge point are memory- or network-bound, while those above are compute-bound. For instance, in Figure 5(a), the INT8 kernel 0 ($K_0$) is compute-bound with HBM but memory-bound with GDDR.

| (a) code prompt. | (b) chat prompt. | (c) code generation. | (d) chat generation. |
|---|---|---|---|

Figure 6: The distribution of prompt and generated tokens in Microsoft Azure cloud.

**Request characterization**. In real-world LLM serving clouds like Microsoft Azure, the distribution of inference configurations such as prompt length and generated token count is not uniform. We analyzed public production traces (Microsoft, 2024) from Azure's code completion and conversation services. The prompt length distributions are shown in Figures 6(a) and 6(b), respectively.

Table 1: The comparison of LLMCO$_2$ against prior work.

| advantage | LLMCarbon | nn-Meter | DeepEn | NNLQP | **ours** |
|---|---|---|---|---|---|
| ML-based regression technique | ✗ | ✓ | ✓ | ✓ | ✓ |
| prefill and decode phases | ✗ | ✗ | ✗ | ✗ | ✓ |
| hardware (core, mem, & net) features | ✓ | ✗ | ✗ | ✗ | ✓ |
| tensor parallelism on multi-GPUs | ✗ | ✗ | ✗ | ✗ | ✓ |
| sampling common inference configs | ✗ | ✗ | ✗ | ✗ | ✓ |

Code completion prompts are often longer, with a median length of 1.5K tokens, compared to chat prompts with a 1.02K median, as they include substantial chunks of existing code. Most requests have prompt lengths under 3K tokens. The distributions of generated tokens are in Figures 6(c) and 6(d); the median is 13 tokens for code completion and 129 for chat, with most requests generating under 600 tokens. Major LLM serving clouds, including Azure, use a mixed continuous batching policy (Agrawal et al., 2024), keeping batch sizes typically at $\leq 2$ (Patel et al., 2024).

## 3 RELATED WORK

We compare LLMCO$_2$ with prior work in Table 1. LLMCarbon (Faiz et al., 2024) employs an equation-based approach using FLOPs to estimate LLM inference carbon footprints, resulting in inaccuracies. Other models like nn-Meter (Zhang et al., 2021), DeepEn (Tu et al., 2023), and NNLQP (Liu et al., 2023) use random forests or neural networks to predict latency or energy for CNNs and transformers but treat predictions as a single task, overlooking the autoregressive nature of LLM inferences and failing to distinguish between prefill and decode phases. These methods focus solely on architecture-specific features like input sizes and network structures, relying on brute-force sampling across hardware platforms, yet they omit hardware-specific features such as GPU peak throughput, memory bandwidth, and network bandwidth, resulting in reduced accuracy for unseen hardware configurations. Furthermore, they do not address tensor parallelism across multiple GPUs and uniformly sample inference configurations, neglecting common settings like medium prompt lengths and small batch sizes. In contrast, LLMCO$_2$ separates prefill and decode phases, incorporates hardware-specific features, supports multi-GPU tensor parallelism, and prioritizes frequently occurring inference configurations, achieving more precise carbon footprint predictions.

## 4 LLMCO2

We introduce LLMCO$_2$, an accurate carbon footprint regression model for LLM inferences, as outlined in Figure 7. Firstly, ❶ we propose a novel graph embedding method that encodes a transformer's layer, running on one or more GPUs during inference, as a graph. In the graph, nodes correspond to the kernels within the transformer layer, while edges depict data dependencies between kernels. Each node is annotated with architectural features such as operation count, memory access size, and network transfer size, along with a hardware-

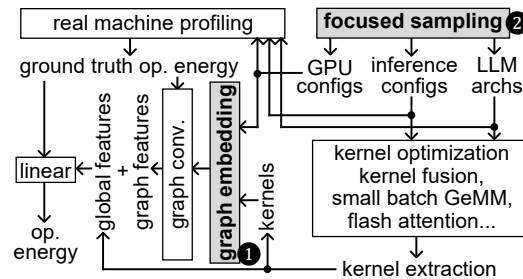

Figure 7: The overview of LLMCO$_2$.

specific feature, i.e., its Roofline performance. Node features are divided into two sets, one for the prefill phase and the other for the decode phase. We then employ graph convolution layers to process this graph. Global LLM architectural features, including total operation count, memory access, and network transfer size, are integrated with the processed graph features, and two linear layers use these to predict the LLM inference's operational energy. To convert operational energy consumption to carbon footprint, we apply the equation from Faiz et al. (2024):

$$CO2eq_{oper} = energy_{oper} \cdot PUE \cdot carb\_inten, \tag{1}$$

where $CO2eq_{oper}$ represents the operational carbon footprint, $PUE$ denotes the power usage effectiveness, and $carb\_inten$ indicates the carbon intensity of the data center. The embedded carbon footprint is calculated using the equations provided in Faiz et al. (2024). Secondly, ❷ we introduce a focused data sampling technique that targets common inference configurations, LLM architectures, and GPU setups frequently encountered by commercial cloud users. This technique generates

Figure 8: Converting the kernels in a transformer layer into a graph.

numerous combinations of inference, LLM architecture, and GPU configurations to build the training dataset. For each combination, the configurations extract kernels in each transformer layer and global LLM features, following various kernel optimizations. These kernels, global LLM features, and hardware configurations form the input data for LLMCO$_2$, while ground truth energy data is gathered from real GPUs, adhering to the specified configurations.

## 4.1 GRAPH EMBEDDING

We introduce a graph embedding technique for LLMCO$_2$ to represent a transformer layer as a directed acyclic graph (DAG) of kernels: $\mathcal{G} = (\mathcal{V}, \mathcal{E})$, where $\mathcal{V}$ is the set of nodes (kernels), and $\mathcal{E}$ denotes the edges (data dependencies between kernels). Each edge's source kernel supplies input data to the destination kernel. The key advantages of this technique are: (1) each node has two distinct feature sets—one for the prefill phase and the other for the decode phase, and (2) each node includes a hardware-specific feature, i.e., Roofline performance. Figure 8 provides an example of this graph embedding for a transformer layer.

**Node features**. A node $v \in \mathcal{V}$ corresponds to a kernel in the transformer layer. To accurately estimate the carbon footprint of the two distinct phases in an LLM inference, each node is assigned two sets of features: one for the prefill phase and the other for the decode phase. Each feature set concatenates the following elements:

- $T$: Defines the type of the kernel. For instance, the value projection kernel $V_{proj}$ multiplies the input with its weight matrix $\mathbf{W}_V$. More kernel types are detailed in Tables 6 and 7 of Appendix A. A one-hot vector is used to encode $T$.
- $S$: Represents the dimensions of the kernel's input, weight, and output.
- $O$: Refers to the number of operations of the kernel. While tools like CALFLOPS (Ye, 2023) can compute operation counts for kernels based on LLM architecture, they do not differentiate between the prefill and decode phases, or support the all-reduce kernel. Therefore, we provide equations for calculating $O$ for the kernels used (excluding the all-reduce kernel) in Appendix A. For the same kernel, the operation count in the decode phase ($O^{dec}$) is proportional to the number of generated tokens ($N_{gT}$), while in the prefill phase ($O^{pre}$), it is proportional to the input sequence length ($L_{seq}$). For the all-reduce kernel operating on an $n \times m$ matrix distributed across $l$ GPUs (each holding an $n/l \times m$ portion), the operation count during the decode phase is given by Equation 2, and during the prefill phase by Equation 3. These operations occur during the reduce-scatter step.

$$O_{allr}^{dec} = n \cdot \frac{m}{l} \cdot (N_{gT} - 1) \qquad (2) \qquad O_{allr}^{pre} = n \cdot \frac{m}{l} \cdot L_{seq} \qquad (3)$$

- $M$: Denotes the total memory footprint accessed by the kernel. We also provide equations for computing $M$ (excluding the all-reduce kernel) in Appendix A. As with $O$, the memory footprint during the decode phase ($M^{dec}$) is proportional to $N_{gT}$, while in the prefill phase ($M^{pre}$), it is proportional to $L_{seq}$. For the all-reduce kernel, the total memory footprint during the decode phase is determined by Equation 4, where $D_A$ represents the data type of activations (e.g., FP16), and in the prefill phase by Equation 5.

$$M_{allr}^{dec} = 2O_{allr}^{dec} \cdot D_A \qquad (4) \qquad M_{allr}^{pre} = 2O_{allr}^{pre} \cdot D_A \qquad (5)$$

- $I$: Indicates the data size transferred over the GPU network interface for an all-reduce kernel. When the all-reduce kernel operates on an $n \times m$ matrix distributed across $l$ GPUs, the data size transferred during the decode phase is calculated using Equation 6, and during the prefill phase using Equation 7.

$$I_{allr}^{dec} = \frac{n}{l} \cdot m \cdot (l-1) \cdot D_A \cdot (N_{gT} - 1) \quad (6) \qquad I_{allr}^{pre} = \frac{n}{l} \cdot m \cdot (l-1) \cdot D_A \cdot L_{seq} \quad (7)$$

- $P$: Denotes the kernel's Roofline performance. The memory-related arithmetic intensity (MAI) for a non-all-reduce kernel is calculated as $O/M$, while the network-related arithmetic intensity (NAI) for an all-reduce kernel is $O/I$. The memory-related ridge point (MRP) of a GPU is computed as $Th_{max}/BW_{max}$, and the network-related ridge point (NRP) as $Th_{max}/NET_{max}$. Here, $Th_{max}$ is the GPU's peak computational throughput for a specific data type (e.g., INT8), $BW_{max}$ is the maximum memory bandwidth, and $NET_{max}$ represents the peak network bandwidth. The kernel's Roofline performance is determined by Equation 8. For non-all-reduce kernels, if the MAI is below the MRP, $P$ equals the product of $BW_{max}$ and the MAI; otherwise, $P$ equals $Th_{max}$. For all-reduce kernels, if the NAI is less than the NRP, $P$ is the product of $NET_{max}$ and the NAI; otherwise, $P$ is $Th_{max}$. Including $P$ as a node feature is essential to account for hardware-specific characteristics, such as $Th_{max}$, $BW_{max}$, and $NET_{max}$, in LLM inference energy modeling.

$$P = \begin{cases} BW_{max} \cdot MAI \, (NET_{max} \cdot NAI) & \text{if } MAI < MRP \, (NAI < NRP) \\ Th_{max} & \text{otherwise} \end{cases} \quad (8)$$

**Global LLM features**. In addition to node features, LLMCO$_2$ incorporates global LLM features, covering the overall architecture of an LLM, batch size, prompt length, generated token count, total FLOP count, memory read/write data size, and network transfer data size. The architectural details include quantization bitwidth, hidden size, intermediate size, head count, and layer count.

**Graph Conv**. We adopted the GraphSAGE layer (Hamilton et al., 2017) for LLMCO$_2$.

## 4.2 Focused Energy Data Sampling

To train a carbon footprint regression model, it is crucial to construct a training dataset by sampling a variety of LLM architectural, inference, and hardware configurations. However, random sampling proves ineffective, often resulting in suboptimal predictive models due to the exclusion of vital data related to LLM architecture and inference configurations. Creating a new LLM demands considerable resources, including substantial training data and computational power. Consequently, most LLMs in academia and industry are based on a few foundational models (Meta, 2023), sharing similar architectural attributes, such as head count, layer number, hidden size, and intermediate size. Neglecting these established architectural parameters and relying on random sampling leads to poor regression accuracy. Additionally, as depicted in Figure 6, the distributions of input prompt lengths and generated token counts are non-uniform in major cloud-based LLM services. Treating all inference configurations uniformly and sampling evenly across all possible inference configurations yields an ineffective training dataset, failing to capture the prevalent configuration ranges of inference requests, thereby compromising the accuracy of carbon footprint predictions.

---

**Algorithm 1:** Focused energy data sampling.

**Input:** $\mathcal{A}$: prior distribution of LLM architectural configs from public repositories; $\mathcal{I}$: prior distribution of inference configs in cloud-based LLM services; $\mathcal{H}$: prior GPU hardware config distribution; TD: initial test dataset; $e$: the error threshold for regression accuracy.

**Output:** Training dataset $(X_{train}, Y_{train})$ and test dataset $(X_{test}, Y_{test})$

1 **def** *FineGraindSampling(X)*:
2     **for** $x \in X$ **do**
3         $D \leftarrow$ randomly sample $B$ data points within a range of $\pm C$ from the original values in $\mathcal{A} \times \mathcal{I} \times \mathcal{H}$
4         $X_{new} \leftarrow X_{new} + D$
5     **end**
6     $Y_{new} \leftarrow$ measure $X_{new}$'s energy on GPU
7     **return** $(X_{new}, Y_{new})$
8 $(X, Y) \leftarrow$ sample $A$ data points from prior distributions $\mathcal{A} \times \mathcal{I} \times \mathcal{H}$.
9 $f \leftarrow$ train the regression model with $(X_{train}, Y_{train})$
10 TD$\leftarrow$TD $+ (X_{test}, Y_{test})$
11 $e(f) \leftarrow$ test $f$ on TD
12 **while** $e(f) > e$ **do**
13     $X^* \leftarrow$ select data points with large error from TD
14     $(X_i, Y_i) \leftarrow FinedGrainedSampling(X^*)$
15     Add $(X_i, Y_i)$ to $(X_{train}, Y_{train})$ or $(X_{test}, Y_{test})$
16     update $f$ with $(X_{train}, Y_{train})$
17     TD$\leftarrow$TD $+ (X_{test}, Y_{test})$
18     $e(f) \leftarrow$ test $f$ on TD
19 **end**

---

**Focused data sampling**. Instead of employing random sampling, we propose a focused energy data sampling strategy, as outlined in Algorithm 1, to identify the most influential data points within the space of LLM architectural, inference, and hardware configurations that significantly affect prediction accuracy. Our approach deliberately omits rarely encountered configurations, such as an LLM with a hidden size of 32 or an inference request generating 16K tokens. Instead, our method itera-

Table 2: The configuration of GPUs used for LLM inferences.

| GPU | max throughput (TOPs/s) | | | memory (GB/s) | network (GB/s) | power ($W$) | node size | area ($mm^2$) | tech (nm) |
|---|---|---|---|---|---|---|---|---|---|
| | FP32 | FP16 | INT8 | | | | | | |
| T4 | 8.1 | 65 | 130 | 320 | 64 | 70 | 4 | 545 | 12 |
| L4 | 121 | 242 | 485 | 300 | 64 | 72 | 4 | 294 | 5 |
| A100 | 312 | 624 | 1248 | 2039 | 600 | 400 | 4 | 826 | 7 |
| H100 | 989 | 1979 | 3958 | 3350 | 900 | 700 | 4 | 814 | 5 |

tively samples additional data around points with high prediction errors. In line 8 of Algorithm 1, we initially select $A = 50K$ data points from the combined space of LLM architecture $\mathcal{A}$, inference $\mathcal{I}$, and hardware $\mathcal{H}$ configurations. We curated $\mathcal{A}$ using 17 state-of-the-art LLMs, adjusting parameters such as head count, layer count, intermediate size, quantization bitwidth, and hidden size. Inference configurations ($\mathcal{I}$) were derived from public LLM inference request traces (Microsoft, 2024), while four GPU configurations (as detailed in Table 2) were considered for $\mathcal{H}$. Our experimental setup is further explained in Section 5.1. To assess the quality of the sampled data, we train a GNN-based predictor and construct a test dataset (lines 9-11), incorporating 20% of the newly sampled data in each iteration. We then perform fine-grained sampling on data points with significant regression errors (lines 1-7). For each data point, we randomly sample $B = 100$ data points within a range of $\pm C$ from their original values in $\mathcal{A} \times \mathcal{I} \times \mathcal{H}$, where $C$ varies for different configurations. For instance, $C = 10$ for input prompt length, $C = 1$ for generated token count, and $C = 1$ for layer number. This iterative process continues until the energy predictor's accuracy reaches the desired target (lines 12-18).

## 5 EVALUATION

### 5.1 EXPERIMENTAL METHODOLOGY

**Dataset construction**. We developed an energy dataset to evaluate the performance of various energy prediction methods, selecting six LLM series: Bloom (Bloom-560m, Bloom-1b1, Bloom-1b7, Bloom-3b, Bloom-7b1) (BigScience, 2023), Gemma (Gemma-2b, Gemma-7b) (Team et al., 2024a), Gemma2 (Gemma2-2b, Gemma2-9b, Gemma2-27b) (Team et al., 2024b), Qwen2 (Qwen2-0.5b, Qwen2-1.5b, Qwen2-7b, Qwen2-72b) (Yang et al., 2024), Llama3.1 (Llama3.1-8b, Llama3.1-70b) (Meta, 2024), and Mixtral (Mixtral-8×7b) (Jiang et al., 2024). The LLM architectural design space was explored by varying parameters such as quantization bitwidth, hidden size, intermediate size, head count, and layer count. Public Azure LLM serving traces (Microsoft, 2024) were used to generate inference requests, encompassing two distinct traces: one for chat (19,336 entries) and the other for code completion (8,199 entries), for LLMCO$_2$. We sampled inference configurations by adjusting input prompt lengths and generated token numbers, ensuring batch sizes remained under two. We adopted random sampling to generate inference requests with different input prompt lengths and generated token numbers for our baseline schemes. LLM inferences were executed using the GPU configurations detailed in Table 2, with the number of GPUs per inference ranging from the minimum required to a maximum of four, regardless of each hardware configuration's total GPU capacity. For the training dataset, we considered L4, A100, and H100 GPUs, while all four GPU configurations were included in the test dataset.

**Measurement and implementation**. We used the Nvidia Management Library (NVML) (NVIDIA, 2024) to measure the energy consumption of LLM inferences on the target GPUs. Each inference was executed 5 times, and the average energy consumption was recorded as the ground truth. LLMCO$_2$ consists of two graph convolution layers and two linear layers. LLMCO$_2$ was developed using the PyG package (Team, 2024) and trained on a Tesla L4 GPU. The model training used the Adam optimizer, with a learning rate of 0.001 and a batch size of 512.

**Evaluation metrics**. We evaluated prediction accuracy using the Mean Absolute Percentage Error (MAPE) and Error Bound Accuracy (EBA($\delta$)). MAPE quantifies the average absolute percentage deviation between predicted and actual energy values, with lower values indicating higher accuracy. EBA($\delta$) represents the percentage of predictions within a specified error bound $\delta$ of the ground truth, with higher values reflecting greater regression precision.

**Baseline Schemes**. To assess the performance of LLMCO$_2$, we compared it against three baseline schemes: LLMCarbon (Faiz et al., 2024), DeepEn (Tu et al., 2023), and NNLQP (Liu et al., 2023).

Table 3: The mean absolute percentage error (MAPE) comparison between various energy predictors. (The lower, the better)

| Scheme | Gemma | Bloom | Gemma2 | Qwen2 | Mixtral | Llama3.1 | Mean |
|--------|-------|-------|--------|-------|---------|----------|------|
| LLMCarbon | 89.2% | 98.5% | 101.2% | 71.3% | 156% | 233% | 124.9% |
| DeepEn | 34.3% | 39.1% | 41.7% | 33.8% | 23.2% | 19.7% | 31.9% |
| NNLQP | 26.8% | 30.6% | 31.6% | 21.5% | 34.1% | 26.2% | 28.5% |
| **LLMCO$_2$** | 15.8% | 11.9% | 21.1% | 6.3% | 19.4% | 18.3% | 15.5% |

LLMCarbon is an equation-based approach that estimates carbon footprint by counting only the FLOPs involved in an LLM inference. DeepEn (Tu et al., 2023) samples the energy consumption of each kernel across different GPUs and utilizes this dataset to train a random forest-based predictor. The only distinction between DeepEn and nn-Meter (Zhang et al., 2021) is that nn-Meter is trained to predict inference latency instead of energy consumption. NNLQP (Liu et al., 2023), a GNN-based energy predictor, represents each model layer as a graph. However, neither DeepEn nor NNLQP distinguishes between the prefill and decode phases of LLM inferences, incorporates hardware-specific features, or emphasizes sampling common LLM architecture, inference, and GPU configurations frequently used in cloud environments.

## 5.2 OPERATIONAL ENERGY RESULTS

**MAPE**. Table 3 presents the comparison of Mean Absolute Percentage Error (MAPE) between LLMCO$_2$ and various baseline schemes. On average, LLMCO$_2$ achieves the lowest MAPE values across different LLMs. LLMCarbon estimates operational energy consumption based solely on FLOP counts, neglecting memory accesses within transformer layers and critical hardware-specific features, such as peak GPU memory and network bandwidth, resulting in the highest MAPE values for all LLMs. The two ML-based predictors, DeepEn and NNLQP, exhibit comparable average MAPE values. Due to a smaller training energy dataset size for Mixtral and Llama3.1, DeepEn, leveraging its random forest model, performs slightly better than NNLQP on these LLMs. Overall, LLMCO$_2$ outperforms DeepEn and NNLQP, reducing the average MAPE by 51.4% and 45.6%, respectively, by treating the prefill and decode phases separately, incorporating kernel-specific Roofline performance, and training with energy data derived from public Azure LLM serving traces.

Table 4: The error bound accuracy (EBA) comparison between various energy predictors. (The higher, the better)

| Metric | Scheme | Gemma | Bloom | Gemma2 | Qwen2 | Mixtral | Llama3.1 | Mean |
|--------|--------|-------|-------|--------|-------|---------|----------|------|
| EBA(30%) | LLMCarbon | 7.8% | 6.1% | 3.8% | 11.9% | 4.2% | 3.5% | 6.2% |
| | DeepEn | 58.5% | 48.9% | 44.8% | 60.5% | 41% | 36.6% | 48.4% |
| | NNLQP | 60% | 50.1% | 45.6% | 65.2% | 37.5% | 33.4% | 48.6% |
| | **LLMCO$_2$** | 77.9% | 85.3% | 71.3% | 80.1% | 64.8% | 62.6% | 73.6% |
| EBA(10%) | LLMCarbon | 2.1% | 1.9% | 0.8% | 7.7% | 1.3% | 0.8% | 2.4% |
| | DeepEn | 25% | 13% | 12.5% | 30.2% | 13.2% | 11.6% | 17.6% |
| | NNLQP | 32.6% | 15.7% | 19.3% | 36.5% | 9.8% | 9.2% | 20.5% |
| | **LLMCO$_2$** | 53.3% | 60% | 40.3% | 53.1% | 35.3% | 32.1% | 45.7% |
| EBA(5%) | LLMCarbon | 0.1% | 0% | 0.1% | 0.3% | 0% | 0% | 1.1% |
| | DeepEn | 11.6% | 8.2% | 4.7% | 16.1% | 7.2% | 6.8% | 9.1% |
| | NNLQP | 11.9% | 13.2% | 7.4% | 19.2% | 6.3% | 6.1% | 10.7% |
| | **LLMCO$_2$** | 27.1% | 46.7% | 12.1% | 33.7% | 9.3% | 7.3% | 22.7% |

**EBA**. Table 4 presents the Error Bound Accuracy (EBA) for various energy predictors at 5%, 10%, and 30% error bounds. LLMCO$_2$ consistently achieves the highest EBA values across all error bounds and LLMs. In contrast, LLMCarbon records the lowest EBA values, as it fails to account for memory accesses and network data transfers, resulting in poor performance, especially with LLMs featuring complex decode phases, such as Gemma2, and those constrained by all-reduce kernels, like Mixtral and Llama3.1. Although DeepEn and NNLQP exhibit similar MAPE values, NNLQP achieves higher EBA values at smaller error bounds due to its more effective GNN model in energy data regression when ample training data is available. Ultimately, LLMCO$_2$ outperforms DeepEn and NNLQP, improving the average EBA(10%) by 160% and 123%, respectively.

Table 5: The EBA(10%) comparison between various components of LLMCO$_2$.

| Scheme | Gemma | Bloom | Gemma2 | Qwen2 | Mixtral | Llama3.1 | Mean |
|---|---|---|---|---|---|---|---|
| +prefill/decode | 41.5% | 38.5% | 29.5% | 44.6% | 26.9% | 24.6% | 34.3% |
| +Roofline | 45.6% | 46.3% | 33.5% | 47.9% | 30.8% | 28.6% | 38.8% |
| +focused sample | 53.3% | 60% | 40.3% | 53.1% | 35.3% | 32.1% | 45.7% |

## 5.3 ABLATION STUDIES

We conducted ablation studies on EBA(10%) to evaluate the contribution of each component of LLMCO$_2$, as summarized in Table 5. By using distinct node features for the prefill and decode phases, LLMCO$_2$ improves EBA(10%) by 67% compared to NNLQP. In real-world LLM serving clouds, most inference requests involve medium-length prompts and fewer generated tokens, leading to significant errors when combining the two phases for carbon overhead prediction. Incorporating Roofline performance as a node feature further boosts LLMCO$_2$'s EBA(10%) by 13.1%, as this feature facilitates knowledge transfer from L4 to T4 GPUs in the test dataset. Finally, the focused energy data sampling technique elevates LLMCO$_2$'s EBA(10%) improvement to 123% of NNLQP, since training with data distributions that mirror real-world LLM inference configurations enhances its performance on test datasets with similar prompt lengths and generated token distributions.

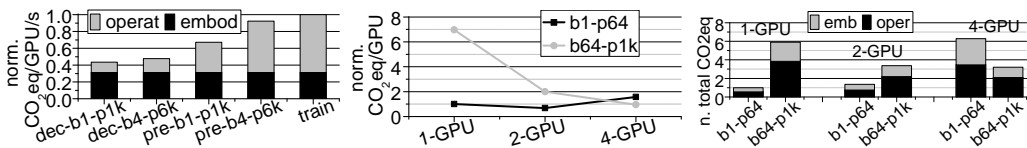

Figure 9: The comparison of carbon per GPU per second.

Figure 10: The oper. carbon per GPU of a Bloom-7b1 inference.

Figure 11: The total carbon of a Bloom-7b1 inference.

## 6 USER CASE STUDIES

**Carbon comparison between training and inference**. LLM training requires numerous GPUs operating at high throughput over extended periods. For example, Mixtral-8×7b training uses 512 A100 GPUs at about 50% peak throughput for three months (Jiang et al., 2024). In contrast, INT8 Mixtral-8×7b inference utilizes four A100 GPUs at 10%–40% peak throughput for around 4 seconds. Figure 10 shows the normalized carbon footprint per GPU per second for Mixtral-8×7b training and inference, relative to the training baseline. The embodied carbon per GPU per second is consistent across all configurations since A100 GPUs are used. The prefill phase of an inference with a batch size of 4 and a 6K prompt length has a similar operational carbon footprint per GPU per second to training, while the decode phase with a batch size of 1 and a 1K prompt length has a much lower footprint. Unlike training, the embodied carbon overhead dominates the decode phase of Mixtral-8×7b inferences.

**Inference on multiple GPUs**. Additional GPUs can accelerate LLM inference (Pope et al., 2023), but using more GPUs than necessary is often unsustainable, particularly for real-world cloud inference configurations with small batch sizes and short prompts. Figure 10 shows the operational carbon footprint of Bloom-7b1 inferences across different GPU counts with varying prompt lengths and batch sizes. For example, although an inference with a batch size of 4 and a 1K-token prompt benefits from 2 or 4 GPUs, lowering the per-GPU carbon footprint, an inference with a batch size of 1 and a 64-token prompt incurs increased latency and higher per-GPU operational carbon due to the communication overhead of all-reduce kernels when using more GPUs. As shown in Figure 11, the total carbon overhead rises considerably with more GPUs for inferences with smaller batch sizes and prompts.

## 7 CONCLUSION

LLM inference produces a larger carbon footprint than training, necessitating accurate estimation tools for both users and cloud providers. Existing models fall short due to their inability to capture LLM autoregressive behaviors, hardware-specific features, and real-world configuration distribution. We presented LLMCO$_2$, a GNN-based model to address these challenges, offering improved accuracy in predicting the carbon footprint of LLM inferences compared to prior methods.

## ETHICS STATEMENT

LLMCO$_2$ offers accurate carbon footprint predictions, helping users and cloud providers make sustainable LLM deployment choices. By enhancing efficiency and transparency, LLMCO$_2$ promotes environmentally responsible AI practices. Its failure would not worsen existing baselines.

## REPRODUCIBILITY STATEMENT

To ensure reproducibility, we provide the complete source code as supplementary material, with detailed instructions for model training, inference, and evaluation included in the paper and Appendix.

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

Figure 12: The kernels in a transformer layer.

## A    THE KERNELS IN A TRANSFORMER LAYER

**Linear kernels**. The kernels in a transformer layer without kernel optimizations are illustrated in Figure 12. The configuration with flash-attention is shown in Figure 12(a), while the configuration without flash-attention is depicted in Figure 12(b). The operation counts and GPU memory usage for all kernels in a transformer layer, with and without flash-attention, are provided in Table 6 and Table 7, respectively. The operation count for all linear kernels during the decode phase ($O_{linear}^{dec}$) is computed as:

$$O_{linear}^{dec} = 2 \cdot |B| \cdot d_{in} \cdot d_{out} \cdot (N_{gT} - 1)/N_{GPU}, \tag{9}$$

where $|B|$ is the batch size, $d_{in}$ is the input dimension, $d_{out}$ is the output dimension, $N_{gT}$ is the number of generated tokens and $N_{GPU}$ is the number of GPUs. During the decode phase, each linear kernel loads weights, the size of which is given by:

$$M_{W,load,linear}^{dec} = d_{in} \cdot d_{out} \cdot D_W \cdot (N_{gT} - 1)/N_{GPU}, \tag{10}$$

where $D_W$ represents the data type of the weights, such as FP16. Each linear kernel reads activation values, the size of which can be calculated as:

$$M_{A,load,linear}^{dec} = d_{in} \cdot |B| \cdot D_A \cdot (N_{gT} - 1)/N_{GPU}, \tag{11}$$

where $D_A$ is the data type of the activation values. Only the $K_{proj}$ and $V_{proj}$ kernels need to store activation values, with the size computed as:

$$M_{A,store,linear}^{dec} = d_{out} \cdot |B| \cdot D_A \cdot (N_{gT} - 1)/N_{GPU}. \tag{12}$$

For other linear kernels, $M_{A,store,linear}^{dec} = 0$. No linear kernel reads from the KV cache, but most types of linear kernels store data in the KV cache, except for the $K_{proj}$ and $V_{proj}$ kernels, where $M_{KV,store,linear}^{dec} = 0$. The size of KV cache data stored by other linear kernels during the decode phase is given by:

$$M_{KV,store,linear}^{dec} = d_{out} \cdot |B| \cdot D_{KV} \cdot (N_{gT} - 1)/N_{GPU}, \tag{13}$$

where $D_{KV}$ represents the data type of the KV cache values. The total memory access for linear kernels during the decode phase can be computed as:

$$M^{dec} = M_{W,load,linear}^{dec} + M_{A,load,linear}^{dec} + M_{A,store,linear}^{dec} + M_{KV,store,linear}^{dec}. \tag{14}$$

In contrast, for all linear kernels, the operation count during the prefill phase ($N_{op,linear}^{pre}$) can be calculated as:

$$N_{linear}^{pre} = 2 \cdot |B| \cdot d_{in} \cdot d_{out} \cdot L_{seq}/N_{GPU}, \tag{15}$$

where $L_{seq}$ represents the input prompt sequence length. During the prefill phase, each linear kernel loads weights, with the size calculated as:

$$M_{W,load,linear} = d_{in} \cdot d_{out} \cdot D_W/N_{GPU}, \tag{16}$$

where $D_W$ is the data type of the weights, such as FP16. The size of activation values for each linear kernel during the prefill phase is computed as:

$$M_{A,load,linear}^{pre} = d_{in} \cdot |B| \cdot D_A \cdot L_{seq}/N_{GPU}. \tag{17}$$

The activation value size in the prefill phase can be computed as:

$$M_{A,store,linear}^{pre} = d_{out} \cdot |B| \cdot D_A \cdot L_{seq}/N_{GPU}. \tag{18}$$

Table 6: The computation, memory access, and network transfer of kernels with flash attention.

| kernel | OPs | | memory | | network | |
|---|---|---|---|---|---|---|
| | prefill | decode | prefill | decode | prefill | decode |
| $norm_{attn}$ | Eq 51 | Eq 43 | Eq 57 | Eq 49 | 0 | 0 |
| $Q_{proj}$ | Eq 15 | Eq 9 | Eq 20 | Eq 14 | 0 | 0 |
| $K_{proj}$ | Eq 15 | Eq 9 | Eq 20 | Eq 14 | 0 | 0 |
| $V_{proj}$ | Eq 15 | Eq 9 | Eq 20 | Eq 14 | 0 | 0 |
| $fuse_{attn}$ | Eq 38 | Eq 27 | Eq 42 | Eq 31 | 0 | 0 |
| $out_{proj}$ | Eq 15 | Eq 9 | Eq 20 | Eq 14 | 0 | 0 |
| $add_{attn}$ | Eq 52 | Eq 44 | Eq 57 | Eq 49 | 0 | 0 |
| $norm_{mlp}$ | Eq 51 | Eq 43 | Eq 57 | Eq 49 | 0 | 0 |
| $gate_{proj}$ | Eq 15 | Eq 9 | Eq 20 | Eq 14 | 0 | 0 |
| $up_{proj}$ | Eq 15 | Eq 9 | Eq 20 | Eq 14 | 0 | 0 |
| $act_{mlp}$ | Eq 53 | Eq 45 | Eq 58 | Eq 50 | 0 | 0 |
| $down_{proj}$ | Eq 15 | Eq 9 | Eq 20 | Eq 14 | 0 | 0 |
| $add_{mlp}$ | Eq 52 | Eq 44 | Eq 57 | Eq 49 | 0 | 0 |
| all reduce | Eq 3 | Eq 2 | Eq 5 | Eq 4 | Eq 7 | Eq 6 |

No linear kernel reads from the KV cache, but most types of linear kernels store data in the KV cache. Only the $K_{proj}$ and $V_{proj}$ kernels do not store data in the KV cache, so $M^{pre}_{A,store,linear} = 0$ for these kernels. The size of data stored in the KV cache by other linear kernels during the prefill phase can be computed as:

$$M^{pre}_{KV,store,linear} = d_{out} \cdot |B| \cdot D_{KV} \cdot L_{seq}/N_{GPU}. \tag{19}$$

The total memory access for linear kernels during the prefill phase can be calculated as:

$$M^{pre} = M_{W,load,linear} + M^{pre}_{A,load,linear} + M^{pre}_{A,store,linear} + M^{pre}_{KV,store,linear}. \tag{20}$$

**Kernels in attention**. The attention head dimension $d_h$ equals the hidden size ($size_h$) divided by the number of attention heads ($n_h$). For a transformer layer with no flash attention, in the decode phase, the operation number of a kernel of $matmul_{SV}$ or $matmul_{QK}$ ($O^{dec}_{matmul}$) can be calculated as

$$O^{dec}_{matmul} = |B| \cdot d_h \cdot n_h \cdot (2L_{seq} + N_{gT}) \cdot N_{gT}/N_{GPU}, \tag{21}$$

where $L_{seq}$ is the input prompt length, while $N_{gT}$ is the number of newly generated tokens. The operation number in a softmax activation in the decode phase can be computed as

$$O^{dec}_{softmax} = 5|B| \cdot n_h \cdot (2L_{seq} + N_{gT}) \cdot N_{gT}/2/N_{GPU}. \tag{22}$$

These kernels do not need to load weights or store data in the KV cache. In the decode phase, the kernel of $matmul_{SV}$, $matmul_{QK}$ or softmax loads or stores activation values, whose size is

$$M^{dec}_{A,load/store,matmul/softmax} = |B| \cdot n_h \cdot D_A \cdot (2L_{seq} + N_{gT}) \cdot N_{gT}/2/N_{GPU}, \tag{23}$$

where $D_A$ is the data type of activations. The kernel of $matmul_{SV}$ or $matmul_{QK}$ also needs to load data from the KV cache and the size of these data can be computed as

$$M^{dec}_{KV,load,matmul} = |B| \cdot d_h \cdot n_{kv} \cdot D_{KV} \cdot (2L_{seq} + N_{gT}) \cdot N_{gT}/2/N_{GPU}, \tag{24}$$

where $n_{KV}$ is the number of key-value heads, while $D_{KV}$ indicates the data type of KV cache data. In the decode phase, the total memory access size of $matmul_{SV}$ or $matmul_{QK}$ can be computed as

$$M^{dec}_{matmul} = M^{dec}_{A,load,matmul} + M^{dec}_{A,store,matmul} + M^{dec}_{KV,load,matmul}, \tag{25}$$

while the total memory access size of softmax is

$$M^{dec}_{softmax} = M^{dec}_{A,load,softmax} + M^{dec}_{A,store,softmax}. \tag{26}$$

For a transformer layer with flash attention, during the decode phase, the operation count for the $fuse_{attn}$ kernel can be calculated as:

$$O^{dec}_{fuse} = 2O^{dec}_{matmul} + O^{dec}_{softmax}. \tag{27}$$

The total size of activation values loaded by $fuse_{attn}$ can be computed as:

$$M_{A,load,fuse}^{dec} = d_h \cdot |B| \cdot n_h \cdot D_A \cdot (N_{gT} - 1)/N_{GPU}. \tag{28}$$

The total size of activation values stored by $fuse_{attn}$ is given by:

$$M_{A,store,fuse}^{dec} = 2d_h \cdot |B| \cdot n_h \cdot D_A \cdot (N_{gT} - 1)/N_{GPU}, \tag{29}$$

The total size of data loaded from the KV cache by $fuse_{attn}$ is:

$$M_{KV,load,fuse}^{dec} = 2|B| \cdot s_{block} \cdot d_h \cdot n_{KV} \cdot D_{KV} \cdot (2L_{seq} + N_{gT}) \cdot N_{gT}/2/N_{GPU}. \tag{30}$$

where $s_{block}$ is the number of KV-heads that can be stored in the GPU on-chip memory, and $n_{KV}$ is the number of key-value heads. The total memory access size of $fuse_{attn}$ during the decode phase is:

$$M_{fuse}^{dec} = M_{A,load,fuse}^{dec} + M_{KV,load,fuse}^{dec} + M_{KV,load,fuse}^{dec} \tag{31}$$

In contrast, during the prefill phase, the operation count for the kernel of $matmul_{SV}$ or $matmul_{QK}$ ($O_{matmul}^{pre}$) can be calculated as:

$$O_{matmul}^{pre} = 2|B| \cdot d_h \cdot n_h \cdot L_{seq}/N_{GPU}. \tag{32}$$

The operation count for a softmax activation in the prefill phase is computed as:

$$O_{softmax}^{pre} = 5|B| \cdot n_h \cdot L_{seq}/N_{GPU}. \tag{33}$$

These kernels do not need to load weights or store data in the KV cache. In the prefill phase, the kernel of $matmul_{SV}$, $matmul_{QK}$, or softmax loads or stores activation values, the size of which is:

$$M_{A,load/store,matmul/softmax}^{pre} = |B| \cdot n_h \cdot D_A \cdot L_{seq}/N_{GPU}, \tag{34}$$

The kernel of $matmul_{SV}$ or $matmul_{QK}$ also loads data from the KV cache, with the size computed as:

$$M_{KV,load,matmul}^{pre} = |B| \cdot d_h \cdot n_{kv} \cdot D_{KV} \cdot L_{seq}/N_{GPU}. \tag{35}$$

In the prefill phase, the total memory access size for $matmul_{SV}$ or $matmul_{QK}$ can be computed as:

$$M_{matmul}^{pre} = M_{A,load,matmul}^{pre} + M_{A,store,matmul}^{pre} + M_{KV,load,matmul}^{pre}, \tag{36}$$

while the total memory access size for softmax is:

$$M_{softmax}^{pre} = M_{A,load,softmax}^{pre} + M_{A,store,softmax}^{pre}. \tag{37}$$

For a transformer layer with flash attention, in the prefill phase, the operation count for the $fuse_{attn}$ kernel can be computed as:

$$O_{fuse}^{pre} = (2O_{matmul}^{pre} + O_{softmax}^{pre}) \cdot L_{seq}. \tag{38}$$

The total size of activation values loaded by $fuse_{attn}$ can be calculated as:

$$M_{A,load,fuse}^{pre} = d_h \cdot |B| \cdot n_h \cdot D_A \cdot L_{seq}/N_{GPU}, \tag{39}$$

while the total size of activation values stored by $fuse_{attn}$ is:

$$M_{A,store,fuse}^{pre} = 2d_h \cdot |B| \cdot n_h \cdot D_A \cdot L_{seq}/N_{GPU}, \tag{40}$$

The total size of data loaded from the KV cache by $fuse_{attn}$ is:

$$M_{KV,load,fuse}^{pre} = 2|B| \cdot s_{block} \cdot d_h \cdot n_{KV} \cdot D_{KV} \cdot L_{seq}/N_{GPU}. \tag{41}$$

The total memory access size for $fuse_{attn}$ during the prefill phase is:

$$M_{fuse}^{pre} = M_{A,load,fuse}^{pre} + M_{KV,load,fuse}^{pre} + M_{KV,load,fuse}^{pre} \tag{42}$$

**normalization**, **residual add**, and **MLP act**. For these layers, no weights are loaded, and there is no access to the KV cache. For each normalization layer during the decode phase, the operation count can be computed as:

$$O_{norm}^{dec} = 7|B| \cdot size_h \cdot (N_{gT} - 1)/N_{GPU}, \tag{43}$$

Table 7: The computation, memory access, and network transfer of kernels without flash attention.

| kernel | OPs | | memory | | network | |
|---|---|---|---|---|---|---|
| | prefill | decode | prefill | decode | prefill | decode |
| $norm_{attn}$ | Eq 51 | Eq 43 | Eq 57 | Eq 49 | 0 | 0 |
| $Q_{proj}$ | Eq 15 | Eq 9 | Eq 20 | Eq 14 | 0 | 0 |
| $K_{proj}$ | Eq 15 | Eq 9 | Eq 20 | Eq 14 | 0 | 0 |
| $V_{proj}$ | Eq 15 | Eq 9 | Eq 20 | Eq 14 | 0 | 0 |
| $matmul_{QK}$ | Eq 32 | Eq 21 | Eq 25 | Eq 36 | 0 | 0 |
| $softmax$ | Eq 33 | Eq 22 | Eq 26 | Eq 37 | 0 | 0 |
| $matmul_{SV}$ | Eq 32 | Eq 21 | Eq 25 | Eq 36 | 0 | 0 |
| $add_{attn}$ | Eq 52 | Eq 44 | Eq 57 | Eq 49 | 0 | 0 |
| $norm_{mlp}$ | Eq 51 | Eq 43 | Eq 57 | Eq 49 | 0 | 0 |
| $gate_{proj}$ | Eq 15 | Eq 9 | Eq 20 | Eq 14 | 0 | 0 |
| $up_{proj}$ | Eq 15 | Eq 9 | Eq 20 | Eq 14 | 0 | 0 |
| $act_{mlp}$ | Eq 53 | Eq 45 | Eq 58 | Eq 50 | 0 | 0 |
| $down_{proj}$ | Eq 15 | Eq 9 | Eq 20 | Eq 14 | 0 | 0 |
| $add_{mlp}$ | Eq 52 | Eq 44 | Eq 57 | Eq 49 | 0 | 0 |
| all reduce | Eq 3 | Eq 2 | Eq 5 | Eq 4 | Eq 7 | Eq 6 |

where $size_h$ represents the hidden size, and $N_{gT}$ is the number of generated tokens. For each residual add layer during the decode phase, the operation count can be computed as:

$$O_{add}^{dec} = |B| \cdot size_h \cdot (N_{gT} - 1)/N_{GPU}. \tag{44}$$

For each MLP activation layer during the decode phase, the operation count is:

$$O_{act}^{dec} = 2|B| \cdot size_h \cdot (N_{gT} - 1)/N_{GPU}. \tag{45}$$

The size of activation values loaded or stored by a normalization or residual add layer during the decode phase can be computed as:

$$M_{A,load/store,norm/add}^{dec} = |B| \cdot size_h \cdot D_A \cdot (N_{gT} - 1)/N_{GPU}. \tag{46}$$

For an MLP activation layer, the size of activation values loaded during the decode phase is:

$$M_{A,load,act}^{dec} = 2|B| \cdot size_h \cdot D_A \cdot (N_{gT} - 1)/N_{GPU}, \tag{47}$$

and the size of activation values stored is:

$$M_{A,store,act}^{dec} = |B| \cdot size_h \cdot D_A \cdot (N_{gT} - 1)/N_{GPU}. \tag{48}$$

Thus, for each normalization or residual add layer, the total memory access size during the decode phase is:

$$M_{norm/add}^{dec} = 2M_{A,load/store,norm/add}^{dec}. \tag{49}$$

For each MLP activation layer, the total memory access size is:

$$M_{act}^{dec} = 3M_{A,load,act}^{dec}. \tag{50}$$

For each normalization layer during the prefill phase, the operation count can be computed as:

$$O_{norm}^{pre} = 7|B| \cdot size_h \cdot L_{seq}/N_{GPU}, \tag{51}$$

For each residual add layer during the prefill phase, the operation count can be computed as:

$$O_{add}^{pre} = |B| \cdot size_h \cdot L_{seq}/N_{GPU}. \tag{52}$$

For each MLP activation layer during the prefill phase, the operation count is:

$$O_{act}^{pre} = 2|B| \cdot size_h \cdot L_{seq}/N_{GPU}. \tag{53}$$

The size of activation values loaded or stored by a normalization or residual add layer during the prefill phase can be computed as:

$$M_{A,load/store,norm/add}^{pre} = |B| \cdot size_h \cdot D_A \cdot L_{seq}/N_{GPU}. \tag{54}$$

For an MLP activation layer, the size of activation values loaded during the prefill phase is:

$$M_{A,load,act}^{pre} = 2|B| \cdot size_h \cdot D_A \cdot L_{seq}/N_{GPU}, \tag{55}$$

and the size of activation values stored is:

$$M_{A,store,act}^{pre} = |B| \cdot size_h \cdot D_A \cdot L_{seq}/N_{GPU}. \tag{56}$$

Thus, for each normalization or residual add layer, the total memory access size during the prefill phase is:

$$M_{norm/add}^{pre} = 2M_{A,load/store,norm/add}^{pre}. \tag{57}$$

For each MLP activation layer, the total memory access size is:

$$M_{act}^{pre} = M_{A,load,act}^{pre} + M_{A,store,act}^{pre}. \tag{58}$$

