# OpenReview forum: "LLMCO2: Advancing Accurate Carbon Footprint Prediction for LLM Inferences"
_ICLR.cc/2025/Conference — ICLR 2025 Conference Withdrawn Submission_

### Official Review · Reviewer_zUnY · 2024-11-01

**Soundness:** 2
**Presentation:** 2
**Contribution:** 2
**Rating:** 3
**Confidence:** 4

**Summary:**

The paper proposes a graph neural network-based model, called LLMCO2, which aims to improve the accuracy of carbon footprint prediction in the LLM inference process.

**Strengths:**

1. The organization of this paper is good.
2. It can improve the accuracy of LLM inference carbon footprint predictions compared to previous methods.

**Weaknesses:**

1. The paper proposes a GNN-based model to predict the carbon footprint of LLM inferences. The integration of graph embedding, data sampling, and the Roofline model does not introduce fundamentally new concepts but rather repurposes established techniques in a novel application context. This approach lacks substantial innovation. The primary contribution is an improvement in prediction accuracy over previous models. However, these improvements are incremental, and the research lacks a clear breakthrough in model efficiency or in introducing a new predictive paradigm.

2. While the model aims to predict the carbon footprint, it does not adequately address the broader trade-offs in energy efficiency versus accuracy. The model's usage of multiple GPUs and its impact on real-world energy savings are not thoroughly discussed.

**Questions:**

The paper’s focused data sampling strategy is likely to cause biases in the model’s predictions. The sampling method omits rarely encountered configurations, which might negatively affect the model's robustness when faced with less common but real-world scenarios, potentially leading to inaccuracies in edge cases.

---

### Official Review · Reviewer_LWKb · 2024-11-02

**Soundness:** 2
**Presentation:** 2
**Contribution:** 1
**Rating:** 1
**Confidence:** 4

**Summary:**

This paper presents LLMCO2, a graph neural network that provides the inference energy consumption cost of LLMs based on the computation graph and deployed hardware of said LLM. They account for the prefill- and decode-phase of the LLM, utilize hardware characteristics, and keep tensor parallelism in mind, which limits the mean absolute error percentage compared to previous state-of-the-art to ~ 20% w.r.t. ground truth.

**Strengths:**

S1: The authors found a research gap that the previous SOTA did not address.

S2: This is an important topic to work on, and I am glad it is being tackled from a solid basis in empirical research (performing a roofline analysis, considering multiple models, etc.).

S3: An interesting approach to use the computation graph as input to a GNN. I feel there is a lot of value in this following this further in this line of research.

**Weaknesses:**

W1: I seriously doubt the credibility of the papers cited in this line of research, like Faiz et al., which is an important building block in the argumentation of this paper (see first and second paragraph). They cite Strubell et al. 2019, which was famously (or at least I thought) debunked by Patterson et al. (https://arxiv.org/pdf/2104.10350) as providing an 88x higher estimate of the energy consumption compared to what was really used, highlighting how hard it is to perform this research in practice. Patterson et al. was also not cited, which, from my understanding, is one of the few papers of people with actual end-to-end access to all production metrics, making this more trustworthy than most other papers that estimate costs. I take specific issue with general statements "(...) with a single epoch requiring three times the FLOPs of an inference", "(...) the period required for inference emissions to match training emissions is rapidly decreasing." and "For instance, training the Google T5 LLM generates 40% more carbon emissions than a round-trip flight between San Francisco and New York (Faiz et al., 2024)." While the first two statements would have to have a large caveat of the specific use-case attached to it and potentially not being representative at the current time, the last statement is, to my understanding, entirely wrong. Patterson et al. states that T5 training took 46.7 tCO2e (Table 4), which is 26% of a roundtrip between NYC and SF (180 tCO2e). The authors argue that this took 140%, an error by a factor of 140/26 ~= 5x.

W2: I question the addition of empirical data from older hardware towards the training dataset of LLMCO2. Inference for frontier models like OpenAI's (which was used in the motivation) likely does not happen on older GPU architectures due to their missing support of sparsity and reduced FLOP and memory bandwidth per $ performance, making that analysis void. When arguing that inference is costly at scale, one needs to analyze the actual costs of said scale and not use older hardware for that comparison.

W3: The authors cite the Azure LLM trace as a basis for their decision to use a <2 batch size and argue that this is representative. The same paper (Patel et al. '24) also says this only counts for the prefill (Sec 3.D) but not for decoding, which scales for that particular model-hardware setup until a batch size of 64 (only limited by memory in that specific instance, making the results even more questionable as this is dependent on the context lengths of the requests due to them affecting the KV-cache). This is not addressed by LLMCO2, making its prediction very likely not applicable to other providers and use cases. Additionally, Patel et al. argue that the largest latency (hence, time and energy consumption at ~50% energy consumption) comes from the decoding phase, so they split these two phases to run on different hardware setups. However, the authors do not consider this use case (at least to my understanding) when generating the dataset to train LLMCO2, discounting their own motivation of "Disregard for prevalent configurations", while Patel et al. was used to argue this exact motivation.

W4: The idle time of the nodes was not considered, which is also likely to be a large chunk of energy spent if services are not utilized to capacity. Arguably, this can be omitted by assuming that on-demand up-/downscaling of hardware works efficiently. However, this was never stated. Generally, the assumptions of when LLMCO2 will likely provide accurate results and when it does not are not stated concretely enough.

W5: While I appreciate the empirical-based approach of this paper, it is lacking on multiple fronts. No SOTA deployment methods are discussed (while citing Patel et al., which proposes one instance when splitting prefill and decoding phase on different hardware generations), and continuous batching was barely addressed in how it affects energy consumption. No sparsity, no speculative decoding or streaming, no TensorRT, no NVIDIA Triton, no vLLM, no torch.compile, and a very limited definition of workloads. Just as an example, if speculative decoding improves the decoding phase by 2x, its carbon emissions will be approximately halved (due to the smaller model being multiple magnitudes faster and cheaper to run, making it a negligible cost). This is a very common technique and is likely to be used by most inference providers, which is not addressed at all.

W6: Peak FLOP is wrong for the H100 in Table 2 (see the H100 datasheet https://resources.nvidia.com/en-us-tensor-core/nvidia-tensor-core-gpu-datasheet). FP32 is stated as 989 TFLOP, while it is 67 TFLOP (presumably, the authors mistyped TF32, which would also be wrong as this is the performance with sparsity). FP16 performance is stated as 1979 TFLOP, but this is 989 TFLOP, as this is the performance with sparsity enabled. Same thing for INT8. Same issue for all precisions for the A100. This makes me question the other results from this paper if such an important consideration was overlooked.

W7: The second paragraph states, "(...) the period required for inference emissions to match training emissions is rapidly decreasing." While I agree that this can be the case, the way this is argued is not conclusive. We see a trend that compute utilization is doubling every 6 months (https://epochai.org/trends#compute). To argue the point of inference costs becoming closer to the training costs, the same analysis would need to happen for inference (presumably with AI usage numbers, which are likely to be kept private at large).  However, due to the issues from W5, the trace used in this paper would not be representative of the real world, making it probably impossible to argue outside of specific deployments.

W8: Eq. 1, "energy_per_operation x PUE x carb_intensity." I doubt that this is the correct way to estimate energy consumption. This excludes the energy for the nodes itself, including networking, PC internals, and cooling. A DGX H100 node uses roughly 10.2kW, while 8xH100 use 8x700W=5.6kW, making your resulting estimate off by a factor of 2x (https://docs.nvidia.com/dgx/dgxh100-user-guide/introduction-to-dgxh100.html)

W9: What were the exact dataset splits? The dataset and evaluation sections seem to suggest that all evaluated models were part of the training dataset, making me afraid of an overfit happening.

W10 (Summary and final notes): While the basic premise of the paper is interesting to use a computational-graph analysis and using a GNN to process it, the application of this model would lead to the following problems:
- A misguided understanding of how energy costs come about from LLM inference due to very limited real-world application scenarios
- Potentially misleading results about being energy-efficient in theory or inference being much more grave w.r.t. energy-consumption than training. It is important to note that I am not averse to thinking this might be the case, but how it is stated here is definitely wrong. I fear this work being misunderstood and misused, similar to Strubell et al. (and how prior cited work like Faiz et al. was used similarly by the authors).

Minor issues:
- Figures 9, 10, and 11 are squished and are hardly readable.
- Figure 10 was referenced when the authors wanted to reference Figure 11 in " Figure 10 shows the operational carbon footprint of Bloom-7b1 (...)".
- Figure 11: It is hard to understand what training is and what is inference due to the legend's labels.

**Questions:**

Q1: Given the discrepancies in the cited data (as outlined in W1, W2, and W3), how do you plan to reassess and strengthen the motivation for your research?

Q2: Regarding the issues raised in W4, W6, W7, and W8, could you elaborate on your decision-making process for including or excluding certain factors in your model? What were the trade-offs you considered, and how do you plan to address these concerns in future iterations of your work?

Q3: Why were real-world deployment types outlined in W5 not included? The results from your model are not representative if even one of the techniques is used, potentially making the model void outside of the exact specifications under which it was developed in this paper. My current understanding points me towards a mathematical model based on empirical measurements rather than a regression-based predictor for energy consumption. Even if the mathematical model might not be perfectly accurate, I am fairly confident in estimating a tight lower and upper bound of energy consumption with any of these techniques used.

---

### Official Review · Reviewer_cevv · 2024-11-04

**Soundness:** 3
**Presentation:** 3
**Contribution:** 2
**Rating:** 6
**Confidence:** 1

**Summary:**

This paper introduces LLMCO$_2$, a GNN-based pipeline to estimate the carbon footprint of LLM inference.
LLMCO$_2$, separate LLM's prefill and decode stage during inference, and uses focused sampling targeting common inference configurations. The system demonstrates significant improvement in prediction accuracy compared to existing methods across various LLM architectures and GPU configurations.

**Strengths:**

- Addresses the critical issue of carbon footprint estimation for LLM inferences
- This paper separate modeling of prefill/decode phases and provides more accurate estimate
- Use GNN to predict the carbon footprint and shows promising accuracy.

**Weaknesses:**

- This is good work. However,  this paper is more suitable for an HPC-related conference; I didn't see much relation to the submitted track of `alignment, fairness, safety, privacy, and societal considerations'.
- While the paper presents several innovations, it looks like combining existing methods such as GNNs, the Roofline model, and active learning. It may lack of novelty.

**Questions:**

-  Could you provide more details about the exact training setup of the GNN, specifically: what is the target label during training, how is it measured and at what granularity, and how do you handle the separation of prefill and decode phases in your ground truth measurements?

---

### Note · Authors · 2024-11-26

I have read and agree with the venue's withdrawal policy on behalf of myself and my co-authors.